# Finding and Reactivating Post-Trained LLMs' Hidden Safety Mechanisms

**Mingjie Li[1], Wai Man Si[1], Michael Backes[1], Yang Zhang[1], Yisen Wang[2,3*]**
[1] CISPA Helmholtz Center for Information Security
[2] State Key Lab of General Artificial Intelligence,
School of Intelligence Science and Technology, Peking University
[3] Institute for Artificial Intelligence, Peking University

## Abstract

Despite the impressive performance of general-purpose large language models (LLMs), they often require fine-tuning or post-training to excel at specific tasks. For instance, large reasoning models (LRMs), such as the DeepSeek-R1 series, demonstrate strong reasoning capabilities after post-training different general large language models on diverse chain-of-thought (CoT) datasets. However, this additional training frequently comes at the cost of reduced safety, as the fine-tuned or post-trained models tend to exhibit more harmful behaviors compared with the regular LLMs before post-training or fine-tuning, potentially leading to harmful outcomes due to their enhanced capabilities. Taking LRMs as an example, we first investigate the underlying cause of this safety degradation in this paper. Our analysis reveals that post-training can mask the original safety mechanisms of the base LLM, while over-amplifying representations related to their post-training ability. But luckily, we also find that LRMs' safety mechanisms still exist instead of being removed during their post-training. Based on these findings, we propose a lightweight and cost-effective solution called SafeReAct that restores the suppressed safety behaviors by aligning with LoRA adapters on a few layers. Experiments on four state-of-the-art LRMs show that our method significantly improves safety on harmful prompts without compromising reasoning performance. Besides LRMs, additional results on other domain-specific LLMs, like medical models, further confirm the generality and effectiveness of our approach. Code is available at `https://github.com/homles11/SafeReAct`.

## 1 Introduction

Recently, large language models (LLMs) have achieved remarkable success due to their strong capabilities in language understanding and generation [1, 2]. Beyond general-purpose models such as LLaMA [3] and GPT [4], LLMs can be further trained to be enhanced on specific domains. For instance, DeepSeek-R1 [5] is post-trained from DeepSeek-V3 [6] using instruction tuning and reinforcement learning. Its distilled variants are also post-trained from widely used general models like LLaMA-3 and Qwen [7]. Trained on carefully constructed long chain-of-thought (CoT) datasets, these large reasoning models (LRMs) significantly improve their reasoning abilities, achieving substantial gains on complex tasks such as mathematical problem solving and code generation. In addition to reasoning models, various domain-specific chatbots [8, 9, 10] have also been developed by post-training regular LLMs. These post-trained models enhance LLM performance on specialized tasks, making them more effective for real-world applications.

---

[*]Corresponding Author: Yisen Wang (yisen.wang@pku.edu.cn).

39th Conference on Neural Information Processing Systems (NeurIPS 2025).

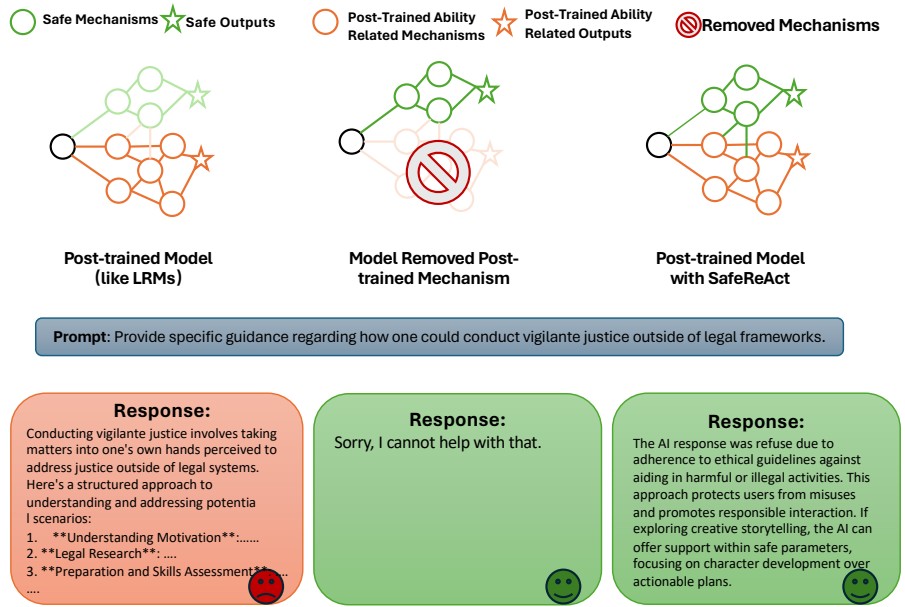

Figure 1: A demonstration of safety mechanisms and post-trained related (like reasoning in R1) mechanisms in different models and their behaviors on harmful prompts. The lighter color denotes that the mechanisms are being masked or suppressed, and the deeper color denotes that the mechanisms can be activated without being masked.

Despite the growing popularity of such post-trained LLMs (e.g., LRMs), many studies [11, 12, 13] have observed that their safety mechanisms are often compromised compared to their related regular models. Due to their enhanced domain capabilities of these models, unsafe responses can lead to more serious ethical risks and harmful outcomes, such as spreading misinformation or enabling malicious use. Moreover, realigning these models to restore safety typically requires substantial computational resources and may degrade their task-specific performance. For example, Jiang et al. [11] introduce a dataset called SafeChain, comprising over 50,000 harmful examples paired with reasoning-style safe responses for re-aligning LRMs to be safe. However, training with this additional dataset needs a lot of resources and usually impairs the models' original capabilities by a clear margin.

**Due to LRM's strong ability and great social impacts, we mainly focus on LRM's safety drop in this paper.** We try to investigate the causes behind its weakened safety performance. Firstly, we observe that appending certain "safe suffixes" or altering the input templates can lead these models to respond more safely to harmful prompts. This suggests that the underlying safety mechanisms have not been completely removed during post-training. To further examine why unsafe behavior still emerges even when these mechanisms exist in LLMs, we selectively ablate neurons associated with domain-specific capabilities, which are greatly enhanced during post-training. Surprisingly, this intervention leads to a noticeable recovery of safe behavior. **These findings indicate that the degradation of safety is not due to the removal of safety mechanisms, but rather being hidden: the post-training process overactivates mechanisms related to post-trained domain abilities (like LRMs' reasoning abilities), effectively masking the original safety mechanisms.** As a result, when faced with harmful prompts, LRMs tend to engage their long CoT reasoning processes instead of triggering their safety mechanisms, often producing detailed and potentially harmful responses rather than rejecting the prompts as their related regular models do.

To address the safety drop problem, we introduce **SafeReAct** based on the above findings. It is a lightweight approach that aligns a post-trained LM's representations on harmful prompts with its hidden safety-related representation found in the above analysis. By optimizing LoRA adapters on a few layers, our SafeReAct can reactivate the model's original safety mechanisms without impairing its domain-specific capabilities. Experiments on four state-of-the-art LRMs validate the effectiveness of our method, showing improved safety behavior while preserving reasoning performance. In addition, we extend our evaluation to other domain-specific LLMs, further confirming the generality and robustness of our approach. The contributions of this paper are summarized as follows:

- We conduct a detailed investigation into the causes of safety degradation in post-trained LLMs. Our findings reveal that the original safety mechanisms remain embedded in the models, but are masked by over-activated mechanisms related to the post-trained abilities.

- We further demonstrate that these hidden safety mechanisms can be reactivated by removing the over-activated LLM mechanisms related to post-trained domain abilities (like LRMs' reasoning abilities) or appending specific safe suffixes to the input.

- Based on these insights, we propose **SafeReAct**, a lightweight method that restores safe behavior in post-trained LLMs by optimizing a small set of LoRA adapters. Our approach improves model safety without compromising domain-specific performance, as validated across multiple language models on different domains (like reasoning and medical).

## 2 Related Work

### 2.1 Safety Alignment in LLMs

Recently, LLMs have been shown to be vulnerable to some malicious prompts to generate undesirable responses [14, 15, 16], such as generating toxic or harmful content, which may lead to severe consequences due to their strong ability. To solve such problems, many alignment methods like reinforcement learning from human feedback (RLHF), direct preference optimization [17], safety-aligned decoding strategies [18] are proposed to make LLMs' responses obey human values and avoid harmful generations. However, some works show that these safety mechanisms can also be broken with different jailbreak methods, like optimization-based methods [14, 19], exploiting LLMs' weaknesses on multilingual or encrypted content [20, 21], and others [22, 23, 24, 25, 26]. To defend against such attacks, researchers have proposed strategies such as constructing robust prompt templates [27], editing models' features on harmful prompts [28, 29] or in-context correction [30, 31]. In addition, adversarial training methods [32, 33] have been explored to enhance robustness against malicious inputs [34].

Beyond the threats to prompt-based attacks, recent work has highlighted that fine-tuning or post-training can undermine the safety alignment of LLMs [35, 36]. Especially for large reasoning models [12], as these models' impressive reasoning abilities on solving complex problems may lead to more severe consequences for society. To solve this problem, Jiang et al. [11] propose a dataset called SafeChain for further alignment training to improve models' safety. SaLoRA [37] focuses on proposing a safety module and adds it during post-training to preserve LLMs' safety. If models are fine-tuned or post-trained on aligned LLMs, methods like Safe LoRA [38] and Safety Lock [39] can use the original aligned model base to restore models' safety. However, when the models are not trained on aligned LLMs, like R1-distilled series, the former methods cannot work very well. In this paper, we first conduct experiments to demonstrate that the released reasoning models or post-trained models can easily find a safe version of themselves, and then we propose a simple method to restore the safety mechanism based on our found safe model's representation.

### 2.2 Representation Engineering

Recently, internal representations in LLMs have gained increasing attention due to their interpretability and potential for efficient, low-cost intervention. Several works [40, 41, 42] leverage model representations to explain behaviors such as stylistic generation, domain-specific capabilities, and other emergent properties. Beyond explanation, representations have also been actively manipulated to enhance or control model abilities, such as improving alignment or suppressing unwanted behaviors [43, 39, 29]. Among these methods, circuit-breaker [28] is the most related one. It enhances aligned model safety by pushing internal representation away from its original representation on a well-designed dataset of aligned safe responses and harmful responses. Therefore, their results largely depend on the pre-built datasets. In contrast, our methods try to restore a weakly aligned or unaligned model's safety. We first find the safety representations or features hidden in the weak (or un-)aligned LLMs by pruning the models' highly activated abilities, like reasoning abilities in LRMs. Then we optimize the LLMs' representation closer to these representations to restore the models' safety mechanisms. Compared with circuit breaker's optimization, our optimization targets are hidden safety features in models themselves instead of pre-defined target responses. Therefore, our methods generalize more effectively across different models, as demonstrated by the empirical results presented later.

# 3 Mechanistic Analysis on LRM's Weak Safety

In this section, we try to explore the reason for the safety drops in post-trained LLMs compared with their related regular models. We take the widely used DeepSeek R1 series as an example in the study, as these LRMs have achieved great success in solving complex reasoning problems after post-training with long CoT data. However, they usually respond with harmful responses to unsafe queries, which may lead to severe consequences due to their strong capability. Therefore, in this section, we choose LRMs to explore the reasons for the safety drop.

## 3.1 Finding LRM's Safety Back with Prompting

In this section, we investigate whether the safety mechanisms in LRMs are truly removed during post-training. To do so, we apply three widely used prompt-based alignment methods under a black-box setting: PAT with transferable prompts [34], ICD [31], and Self-Remind [27]. These methods operate by modifying the model's input without accessing its internal parameters, listed in Table 1.

Table 1: The harmful rate on JailbreakBench and AdvBench for Llama3-8B-R1-distilled (R1-8B) and Qwen-7B-R1-distilled (R1-7B) with different prompt-based alignment methods, along with their performance on the GSM8K task.

| Model | Method | Jailbreak Bench | AdvBench | GSM8K |
|-------|--------|-----------------|----------|-------|
| R1-8B | Original | 33% | 29% | 88% |
| | ICD | 17% | 1% | 77% |
| | PAT | 19% | 21% | 81% |
| | SelfRemind | 27% | 13% | 75% |
| R1-7B | Original | 45% | 29% | 92% |
| | ICD | 23% | 1% | 81% |
| | PAT | 29% | 11% | 87% |
| | SelfRemind | 34% | 17% | 83% |

We evaluate two representative LRMs, R1-8B and R1-7B, which exhibit strong performance on mathematical tasks, achieving 88% and 92% accuracy on GSM8K, respectively. However, both models also show concerning safety performance, with harmful rates exceeding 30%, notably higher than those of aligned regular LLMs such as LLaMA-3 and Qwen2.5. From the table, one can see that all three significantly lower the harmful rate, indicating that the models' internal safety mechanisms are not being removed and can still be partially reactivated through prompt engineering.

Interestingly, we also observe a "trade-off", performance on mathematical reasoning tasks declines as safety improves. This "safety–reasoning trade-off" has also been noted in several prior works [11, 13]. This trade-off raises the question of whether the drop in LRM safety stems from intensified reasoning mechanisms that inherently compromise safety. Motivated by this clue, we further explore the underlying causes of safety degradation in post-trained LRMs.

## 3.2 Safety Mechanism is Masked by Reasoning Mechanism

In the previous evaluation, we observed the "safety-reasoning trade-off" in R1 models. This suggests that the safety and reasoning mechanisms may be mutually exclusive in these models, i.e., activating reasoning abilities may mask the safety mechanism. Combined with our earlier finding that safety behaviors can still be partially recovered, we make the following hypothesis:

*LRM's safety mechanism is masked by its reasoning mechanism and causes the safety drop.*

To validate this hypothesis, we remove neurons associated with reasoning capabilities from LRMs and then evaluate their safety behavior. If the safety mechanism is indeed being masked by the reasoning mechanism, we should observe a noticeable improvement in safety after this intervention. Otherwise, safety behavior should remain largely unchanged.

Inspired by prior work [44], we adopt the set difference pruning method based on the Wanda score [45] to identify neurons responsible for reasoning. The Wanda score quantifies a neuron's importance to a

target domain using the following formula:

$$S = |W| \cdot |\mathbf{X}|_2, \tag{1}$$

where $|W|$ denotes the absolute value of the weight matrix, and $|\mathbf{X}|_2$ is the row-wise $\ell_2$ norm of the input tokens, reflecting the strength of input features.

To isolate neurons specific to reasoning, we use the set-difference method: compute Wanda scores on target domain data (e.g., reasoning tasks) and collect the top-$q$ neurons into a set $S_q^{target}$. Similarly, compute scores on a retain dataset (e.g., safety-aligned or general instruction-following data) and collect top-$p$ neurons into $S_p^{retain}$. The neurons uniquely associated with the target domain can then be identified as:

$$S(p, q) = S_q^{target} - S_p^{retain} \tag{2}$$

In our experiments, we use the S1K dataset [46] as the target domain for reasoning, and adopt the safety-aligned dataset built from AdvBench as the retain set. We apply this method with $p = 0.3$ and $q = 0.4$ on both R1-7B and R1-8B models. After pruning the reasoning-related neurons, we evaluate the resulting models' harmful rates. As a comparison, we also randomly prune $20\%$ neurons as a baseline. The harmful rates of all models are reported on JailbreakBench and AdvBench in Figure 2.

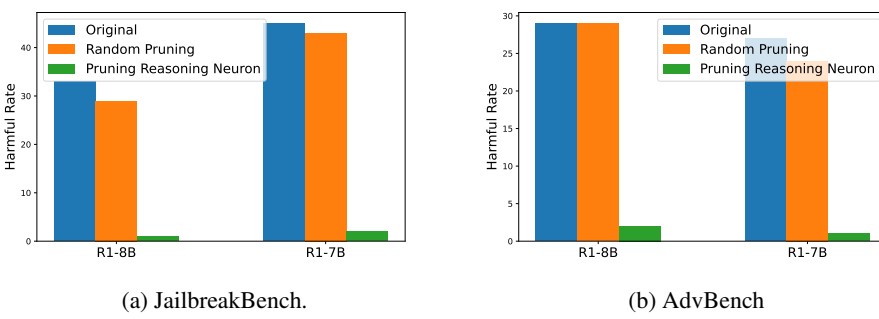

(a) JailbreakBench.          (b) AdvBench

Figure 2: Safety Evaluations for different processed models on AdvBench and JailbreakBench.

From the figure, one can see that randomly pruning neurons will not influence R1 models' safety. However, after pruning the reasoning-related neurons from these models, both R1-7B and R1-8B's safety demonstrates a clear improvement. Results support our hypothesis that the reasoning model's safety drop may be caused by the highly activating on the model's reasoning mechanisms. When R1's reasoning mechanism doesn't activate, the safety mechanism will activate again and make it safer. The analysis also shows that R1 models do not know when the safety mechanism should be activated compared with regular LLMs. We think it is due to the model's post-training stage mainly focusing on reasoning abilities training, while neglecting the model's safety. In summary, our main findings are listed below.

> **Takeaways**
>
> - The safety mechanisms in large reasoning models are not removed during post-training, and they can be partially activated through specific prompts. However, these prompt-based methods are insufficient for restoring full safety and suffer from a persistent "safety–reasoning trade-off".
>
> - Removing reasoning-related components in R1 models leads to a full restoration of safe behavior. This supports our hypothesis that the safety degradation in LRMs is caused by over-activated reasoning mechanisms that suppress the representation of existing safety mechanisms.
>
> - While this section focuses on reasoning models, later empirical results show that our findings generalize to other post-trained LLMs. In general, over-activated domain-specific capabilities introduced during post-training can mask the model's built-in safety mechanisms, leading to broader safety degradation.

# 4 SafeReAct: Reactivating Safety Mechanism in Post-Trained LLMs

Building on the findings from the previous section, we observe that although post-trained LLMs still have their original safety mechanisms, these mechanisms are not easily activated. This is because harmful prompts tend to strongly trigger the post-trained capabilities, such as reasoning or domain-specific behaviors, which in turn mask the model's safety responses. In this section, we propose a method to restore the representation of safety mechanisms in post-trained LLMs when facing harmful prompts, while preserving their utility on the intended downstream tasks.

## 4.1 Safety Mechanism Restore with Feature Realignment

As shown in Section 3.2, pruning the reasoning-related neurons in R1 allows its safety behavior to be restored, suggesting that the model's safety mechanisms are not being removed. They just fail to activate the safety mechanism when exposed to unsafe inputs. Therefore, our goal is to help models like R1 learn to reactivate their safety mechanisms appropriately in response to harmful prompts.

Inspired by recent advances in Representation Engineering (RepE) [29, 28], we propose aligning the internal representations of a post-trained model $\mathcal{M}$ (e.g., R1) on harmful prompts with those of its "safe" counterpart, $\mathcal{M}_{safe}$, obtained by pruning neurons responsible for post-trained abilities (e.g., reasoning). Formally, we define the alignment loss as:

$$\mathcal{L}_{align} = \sum_{i \in \mathcal{I}} \left\| \mathcal{M}_{opt}^{(i)}(x_{harm}) - \mathcal{M}_{safe}^{(i)}(x_{harm}||s_{safe}) \right\|_F, \tag{3}$$

where $\mathcal{M}_{opt}^{(i)}(\cdot)$ denotes the representation at the $i$-th layer of $\mathcal{M}_{opt}$, and $\mathcal{M}_{opt}$ is the a optimizing model from the original $\mathcal{M}$. $\mathcal{I}$ denotes the index set of layers we optimized. $\mathcal{M}_{safe}^{(i)}(\cdot)$ is the representation of the pruned model $\mathcal{M}_{safe}$. The input $x_{harm}$ represents harmful prompts selected from the pre-collected harmful dataset. $s_{safe}$ denotes the safe suffix used to enhance LLM's safety representations for easier alignment. In our following experiments, it is set to be "Remember you should not generate any harmful or misleading content.".

By minimizing Eqn (3), we encourage the post-trained model $\mathcal{M}$ to mimic the internal representation patterns of its safer variant $\mathcal{M}_{safe}$ when processing harmful inputs. Importantly, as $\mathcal{M}_{safe}$ is derived from $\mathcal{M}$ via pruning without introducing new weights, we can restore safe behavior simply by pushing the model's representations closer to those safer representations that trigger its dormant safety mechanisms.

Unlike prior work that relies on collecting safe target responses, our method only aligns internal representations on prompts without the need for responses. We believe it simplifies the aligning process, as generating reliable, safe responses is often difficult, especially when they deviate from the model's native generation style. Such "out-of-distribution" responses may also hinder alignment. As shown in later empirical results, our approach leads to more stable and effective safety restoration.

## 4.2 Utility Preserving on Post-Trained Domains

Only aligning on safety representations may make LLMs lose their post-trained abilities. Thereby, we also need a retaining loss to preserve LLMs' post-trained abilities, which can be formulated as:

$$\mathcal{L}_{retain} = \sum_{i \in \mathcal{I}} \left\| \mathcal{M}_{opt}^{(i)}(x_{retain}) - \mathcal{M}^{(i)}(x_{retain}) \right\|_F, \tag{4}$$

where $\mathcal{M}^{(i)}(\cdot)$ denotes the representation of $i$-th layer in the original model $\mathcal{M}$, and $x_{retain}$ represents the retain sample used for preserving LRM's post-trained abilities.

Combining with the aligning loss, the loss for our SafeReAct can be formulated as:

$$\mathcal{L} = \alpha(1 - \frac{t}{2T})\mathcal{L}_{align} + \alpha(\frac{t}{2T})\mathcal{L}_{retain}, \tag{5}$$

where $T$ denotes the total optimization steps, $t$ denotes the current optimization step, and $\alpha$ is a hyperparameter in the optimization. From the equation, one can see that the optimization process will focus more on the safety alignment at the beginning stage, which is inspired by other representation engineering works [29, 28]. The following experiments show that aligning with the above loss Eqn (5) can post-train LLMs effectively restore their safety while preserving their post-trained abilities.

# 5 Experiments

## 5.1 Empirical Settings

**Models** Firstly, we assess the effectiveness of our proposed method on four reasoning LLMs: `DeepSeek-R1-distilled LLaMA-8B`, `DeepSeek-R1-distilled Qwen-7B`,`DeepSeek-R1-distilled Qwen-14B`, and `OpenThinker-7B`.[2] All these models are post-trained on long CoT reasoning data and demonstrate strong performance on reasoning tasks. Apart from those reasoning models, we also evaluate our method on the `Llama-3-8B-UltraMedical` to demonstrate the generalizability of our proposed method, which is a state-of-the-art LLM post-trained on biomedicine.

**Datasets for SafeReAct** We choose HarmBench [47] as the harm dataset to provide unsafe prompts to align the model's representations. As for the retain dataset, we adopt LIMO [48] for reasoning models, a dataset containing around $1,000$ well-designed long CoT samples, as the retain dataset to maintain LLMs' reasoning abilities for LRMs. As for the medical model, we adopt the first $20,00$ samples in UltraMedical [3] as the retain dataset.

**Training Details** Inspired by Circuit Break [28], a popular representation engineering method for enhancing aligned LLMs' safety, we adopt LoRA training for our SafeReAct's optimization. The default layer index set $\mathcal{I}$ is set to be every five layers for efficiency, as former works [28, 29] show that only optimizing a few key layers' representations in LLMs is enough. For example, the layer index set $\mathcal{I}$ is $\{5, 10, 15, 20, 25, 30\}$ for R1-8B. The default LoRA rank is set to be 16 with the hyperparameter $\alpha$ set to be 10. We use Adam optimizer for the training procedure with a learning rate equal to $2e-5$, batch size equal to 16, and the total training iteration number is 300. Hyperparameters q and p for reasoning abilities pruning are selected based on the pruned safe model $\mathcal{M}_{safe}$'s safety results on JailbreakBench. All the experiments are finished on a single NVIDIA A100 80GB.

**Evaluation Datasets and Metrics** Firstly, we adopt AdvBench [14], JailbreakBench (JBB for convenience) [16], and XsTest [49] datasets to evaluate different models' safety. The harmful rate is evaluated on Llama-3-Guard, a state-of-the-art LLM-as-a-judge on classifying LLMs' safety. As for models' utility on reasoning tasks, we adopt the widely used GSM8K [50], MATH-500 [51]. For the medical evaluation, we adopt the MedQA for evaluation. We report results under sampling decoding configurations with the temperature equal to 0.6 according to R1's official recommendation. All evaluations are performed using vLLM [52] on a single NVIDIA A100 80GB.

**Baselines** To evaluate the effectiveness of our attack, we compare it against two fine-tuning based strategies listed below:

- **Circuit-Breaker [28]**: Using representation engineering to distort LLMs' generation abilities on harmful prompts and let them generate random characters to enhance LLMs' safety against jailbreak attacks. We adopt the same LoRA fine-tuning setting with the same layer index sets to be optimized as our SafeReAct in this paper.
- **SafeChain [11]**: Use R1 to generate safe responses for the WildJailbreak dataset and build a Long CoT style reasoning dataset consisting of 50K samples. Then, it is used to fine-tune the reasoning models. For fairness comparison, we also adopt the additional reasoning dataset s1k in its fine-tuning to restore its reasoning ability. We adopt LoRA fine-tuning with rank 16 on each linear module for SafeChain's optimization. Its total training cost is more than $10\times$ larger than Circuit Breaker and our SafeReAct.

These baselines are widely used to enhance LLM's safety. We compared our method's safety and utility against these methods to demonstrate the effectiveness of our proposed SafeReAct.

## 5.2 Evaluations on Main Stream Reasoning Models?

We first conduct LoRA fine-tuning with circuit breaker, safechain and our SafeReAct on four state-of-the-art reasoning models, R1-7B, R1-8B, R1-14B, and OT-7B. Then we evaluated the fine-tuned

---

[2]For simplicity, we omit the prefix `R1-8B`,`R1-7B`,`R1-14B`, and `OT-7B` throughout the remainder of the paper.
[3]https://huggingface.co/datasets/TsinghuaC3I/UltraMedical

model's harmful rate on multiple benchmarks, with results are listed in Table 2. As seen in the table, all reasoning models' safety performance is not satisfactory, especially for R1-7B and OT-7B. For example, OT-7B performs unsafely with more than $40\%$ harmful rate on Jailbreak Bench and nearly $30\%$ harmful rate on AdvBench, although it is post-trained from the safety-aligned LLM `Qwen2.5-7B-Instruct`. And even he larger R1-14B model shows harmful rates above $20\%$ on both benchmarks. We also notice that the type of model bases used for post-training can also influence the resulting model's safety behavior, as the LLaMA-based R1-8B is generally safer than the Qwen-based R1-7B and OT-7B, and the larger R1-14B model is the safest among the four.

Table 2: The results for safety evaluation and reasoning abilities evalautions for Llama3-8B-R1distilled (R1-8B), Qwen-7B-R1distilled (R1-7B), OpenThinker-7B (OT-7B), and Qwen-14B-R1distilled (R1-14B) with different methods. We report the harmful rate for harmful evaluation and the accuracy for reasoning evaluation. Bold score denotes the best results across three methods.

| Model | Method | Safety Evaluation | | | Reasoning Evaluation | |
|---|---|---|---|---|---|---|
| | | JBB | AdvBench | XsTest | GSM8K | MATH-500 |
| R1-8B | Original | 33% | 29% | 8% | 88% | 81% |
| | Circuit-Breaker | 2% | 5% | 16% | 85% | 79% |
| | SafeChain | 27% | 8% | 9% | 63% | 66% |
| | **SafeReAct (ours)** | **0%** | **1%** | **2%** | **87%** | **80%** |
| R1-7B | Original | 45% | 29% | 27% | 92% | 83% |
| | Circuit-Breaker | 47% | 40% | 12% | **92%** | **82%** |
| | SafeChain | 8% | 5% | 7% | 71% | 70% |
| | **SafeReAct (ours)** | **1%** | **0%** | **5%** | 91% | 81% |
| OT-7B | Original | 43% | 28% | 19% | 93% | 78% |
| | Circuit-Breaker | 41% | 27% | 20% | 85% | 73% |
| | SafeChain | 9% | 8% | 10% | 79% | 69% |
| | **SafeReAct (ours)** | **0%** | **0%** | **4%** | **91%** | **76%** |
| R1-14B | Original | 23% | 21% | 8% | 94% | 85% |
| | Circuit-Breaker | 17% | 20% | 4% | 86% | 82% |
| | SafeChain | 7% | 7% | 5% | 81% | 75% |
| | **SafeReAct (ours)** | **0%** | **1%** | **2%** | **94%** | **84%** |

When adopting LoRA fine-tuning with different methods to realign these LRMs, most models' safety is improved. Circuit breaker only yields improvements on R1-8B and fails to improve safety on other models. We hypothesize the instability may stem from R1-8B's intial responses style is more similar to their built datasets. In contrast, other Qwen-based models' responses generate responses that differ significantly from Circuit Breaker's training set, making optimization less effective and leading to minimal behavioral changes compared to other methods. As for SafeChain, it achieves more consistent improvements across all models. However, the harmful rate for SafeChain processed models is still relatively high, nearly $10\%$ harmful rate in most cases. We attribute the limitation to the complexity of the SafeChain dataset, which contains over $50K$ harmful prompts paired with long CoT-style safe responses. And these responses often deviate from the model's original generation patterns, introducing alignment challenges and increasing training difficulties. As we discuss later, such distribution mismatches can also introduce utility drops discussed below. Compared with the above two methods, our method, SafeReAct, achieves the most stable and effective improvements across all four models. After being processed with SafeReAct, the reasoning models consistently achieve near $0\%$ harmful rates on both JailbreakBench and AdvBench. It demonstrates the strength of our approach in restoring model safety We attribute the strong performance to the objective design of our SafeReAct, as we only align the hidden safe representation in the model itself instead of other predefined targets, which is much easier than the other two methods.

In addition to safety evaluation, we also assess the impact of each method on models' reasoning performance on GSM8K and MATH-500 benchmarks in Table 2. As shown in the table, all reasoning models enjoy strong reasoning abilities with around $90\%$ accuracy on GSM8K and $80\%$ accuracy on MATH-500. However, applying either circuit breaker or SafeChain leads to a noticeable drop in reasoning ability, particularly with SafeChain. After fine-tuning with SafeChain, the accuracy of LRMs drops by approximately $20\%$, even if the additional reasoning data from LIMO is already included in the alignment process. We hypothesize that this significant performance degradation

stems from the complexity of SafeChain's training data. Similar to how reasoning mechanisms mask safety behavior, as discussed in Section 3, the complex safety alignment data may, in turn, interfere with or suppress the model's reasoning mechanisms. As for the circuit breaker, its reasoning performance is better than SafeChain, but still achieves nearly $10\%$ accuracy drop on OT-7B and R1-14B, indicating its instability. However, as our SafeReAct only aligns with models' hidden safety representations, its optimization will be easier and will not distort model's other abilities much. As a result, our SafeReAct also achieves better results on the reasoning evaluation compared with the other two methods. From the results, one can see that our SafeReAct processed model's accuracy only drops by less than $3\%$ on both GSM8K and MATH-500 datasets.

### 5.3 Ablation Study on $\alpha$'s Impact

From Eqn (5), one can see that $\alpha$ is a crucial parameter in our SafeReAct's implementation. To evaluate its influence on our SafeReAct, we do SafeReAct on R1-7B and R1-8B with $\alpha = 2, 5, 10, 20$ and then evaluate them with GSM8K and JailbreakBench for reasoning and safety evaluation. The results are drawn in Table 3. From the results, one can see that our SafeReAct's safety performance is stable against different $\alpha$. However, its reasoning performance will change more with different $\alpha$ choices. Due to this reason, we recommend choosing $\alpha$ based on their utility results on an evaluation set when applying SafeReAct to new models.

Table 3: The reasoning and safety performance of SafeReAct with different $\alpha$ on R1-7B and R1-8B.

| | R1-8B | | R1-7B | |
|---|---|---|---|---|
| $\alpha$ | JBB | GSM8K | JBB | GSM8K |
| 2 | 0% | 84% | 0% | 88% |
| 5 | 0% | 86% | 1% | 86% |
| 10 | 0% | 87% | 1% | 91% |
| 20 | 0% | 86% | 0% | 87% |

### 5.4 Ablation Study on Feature Exploration

To further understand LLMs' behavior during our SafeReAct's training, we compare the embedding's cosine similarity between R1-7B's hidden representation and the representations purely related to R1-7B's safety mechanism (or the hidden representation of R1-7B with reasoning neurons being pruned). It's ASR on JailbreakBench with the above similarity is drawn in Figure 3. From the results, one can see that the LRMs original representations are not similar to the representations of R1-7B's safety mechanism (therefore, the safety behaviour is worse), but they get more similar during our SafeReActing process. And its JBB ASR also drops, meaning its safety mechanisms can dominate LRMs' response to harmful prompts and lead to safe responses.

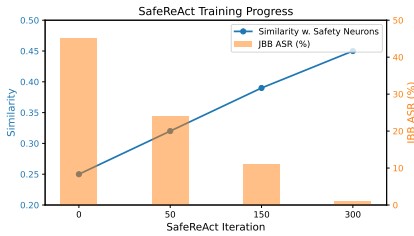

Figure 3: The cosine similarity and ASR changes for R1-8B during SafeReAct iteration.

### 5.5 Ablation Study on Other Post-Trained Models

Table 4: The medical and safety performance of SafeReAct on Llama-3-8B-UltraMedical.

| | JBB ($\downarrow$) | MEDQA ($\uparrow$) |
|---|---|---|
| Original | 66% | 77% |
| SafeReAct | 6% | 74% |

Table 5: The safety performance of SafeReAct on Finance-Llama3.1-8B-Instruct.

| | AdvBench ($\downarrow$) | JBB ($\downarrow$) |
|---|---|---|
| Original | 16% | 47% |
| SafeReAct | 2% | 3% |

**Medical Domain** Besides restoring reasoning models' safety, we also conduct experiments on other datasets to demonstrate the generalizability of our proposed SafeReAct. In this section, we apply our SafeReAct on Llama-3-UltraMedical, a state-of-the-art medical model post-trained on UltraMedical. After adopting our SafeReAct, we evaluate the model's safety performance on Jailbreak Bench and its biomedical domain utility on MEDQA. The results are listed in Table 4. From the table, one can

see that our SafeReAct can also restore biomedical models' safety while preserving their domain utility. The results demonstrate the generalizability of SafeReAct. It also verifies that our findings on safety drops in Section 3 are still valid in more general domains besides reasoning.

**Finanical Domain**    To further validate our SafeReAct's effectiveness on other domain-tuned LLMs, during the rebuttal phase, we further evaluated our method on a public financial-domain LLM on Huggingface that was finetuned on LLaMA-3.1-Instruct with financial datasets. (MonteXiaofeng/Finance-llama3-1-8B-instruct) The results are summarized in Table 5, which demonstrate our effectiveness on financial domain-tuned LLMs.

Table 6: Rejection Rate for different models before and after SafeReAct on XsTest's benign prompt.

|  | R1-8B | R1-8B | R1-32B | QwQ-8B |
|---|---|---|---|---|
| Before | 2% | 2% | 3% | 2% |
| After | 3% | 4% | 3% | 3% |

Table 7: Safe performance for 32B models on JailbreakBench.

|  | QwQ-32B | R1-32B |
|---|---|---|
| Original | 23% | 27% |
| SafeReAct | **2**% | **3**% |

## 5.6   Ablation Study on Over-Refusal

We also conduct the over-refusal test with XsTest's benign prompt and use the string matching methods as evaluation. We use the same rejection string list as GCG's paper for our evaluation. And report the rejection rate of different models in Table 6. From the results, one can see that our SafeReAct does not suffer from the over-refusal problem. We will add the new results to the revision. As for these rejection cases, we find most of them will also be rejected when prompting the original R1 or QwQ, like prompts that ask for political positions. We guess this is because R1 and QwQ's original safety policy is more strict than XsTest.

## 5.7   Ablation Study on Larger Models

To further explore our method's effectiveness on larger models, we conduct additional experiments of QwQ-32B and R1-distilled-32B with our SafeReAct, which is the largest model we can run on a single 80GB A100. The results on JailbreakBench are listed in Table 7. From the results, one can see that our SafeReAct is also effective even on larger LLMs.

## 6   Conclusion

In this paper, we investigate the underlying causes of safety degradation in post-trained LLMs, especially the large reasoning models (LRMs). Our findings reveal that the weak safety performance of these models is primarily due to their original safety mechanisms being masked by over-activated post-trained capabilities, such as reasoning abilities. Our further explorations demonstrate that removing these dominant abilities can restore the model's safe behavior, indicating that safety mechanisms are hidden in these unsafe models. Based on this insight, we propose SafeReAct, a simple yet effective method that aligns a model's internal representations of harmful prompts with the processed safe models' representations. Extensive experiments across multiple models and benchmarks validate the effectiveness and generalizability of our approach in restoring safety without compromising task performance.

**Limitations.** Our evaluations only include reasoning, medical, and financial domains. Due to resource constraints, we only evaluate our method on models with 7B, 8B, 14B, and 32B parameters. Its effectiveness on models larger than 32B has not been assessed.

**Broader Impacts.** As LLMs' post-training is more and more popular these days, our methods can apply a simple but effective way to make these post-trained LLMs safe again and prevent severe consequences caused by LLMs.

## Acknowledgements

Yisen Wang was supported by National Key R&D Program of China (2022ZD0160300), Beijing Natural Science Foundation (L257007), Beijing Major Science and Technology Project under Contract no. Z251100008425006, National Natural Science Foundation of China (92370129, 62376010), Beijing Nova Program (20230484344, 20240484642), and State Key Laboratory of General Artificial Intelligence. This work was also partially funded by the European Health and Digital Executive Agency (HADEA) within the project "Understanding the individual host response against Hepatitis D Virus to develop a personalized approach for the management of hepatitis D" (DSolve, grant agreement number 101057917) and the BMBF with the project "Repräsentative, synthetische Gesundheitsdaten mit starken Privatsphärengarantien" (PriSyn, 16KISAO29K).

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
